# NPEFF: Non-Negative Per-Example Fisher Factorization

## Abstract

As deep learning models are deployed in more and more settings, it becomes increasingly important to be able to understand why they produce a given prediction, but interpretation of these models remains a challenge. In this paper, we introduce a novel interpretability method called NPEFF that is readily applicable to any end-to-end differentiable model. It operates on the principle that processing of a characteristic shared across different examples involves a specific subset of model parameters. We perform NPEFF by decomposing each example's Fisher information matrix as a non-negative sum of components. These components take the form of either non-negative vectors or rank-1 positive semi-definite matrices depending on whether we are using diagonal or low-rank Fisher representations, respectively. For the latter form, we introduce a novel and highly scalable algorithm. We demonstrate that components recovered by NPEFF have interpretable tunings through experiments on language and vision models. Using unique properties of NPEFF's parameter-space representations, we ran extensive experiments to verify that the connections between directions in parameters space and examples recovered by NPEFF actually reflect the model's processing. We further demonstrate NPEFF's ability to uncover the actual processing strategies used by a TRACR-compiled model. We further explore a potential application of NPEFF in uncovering and correcting flawed heuristics used by a model. We release our code to faciliate research using NPEFF.

## 1 Introduction

Neural networks trained on large datasets have achieved human-level performance on many tasks. Unfortunately, these models do not provide any direct method to understand what they have learned or how they have solved the task (Smilkov et al., 2017; Dabkowski & Gal, 2017; Sundararajan et al., 2017). This lack of interpretability can hamper progress in developing new machine learning models and can act as a hindrance to their adoption by making it hard to trust that the model's predictions are based on sound principles (Li et al., 2022).

In this work, we propose NPEFF (**N**on-Negative **P**er-**E**xample **F**isher **F**actorization), an interpretability method that makes use of the model's *parameter space* to produce representations of concepts. Our method can be used with any end-to-end differentiable model without requiring any customization to particular architectures. NPEFF extracts these concepts unsupervisedly given a set of examples. It also provides a theoretically principled way to produce guided changes in a given model's behavior by using these concept representations to directly alter a model's parameters.

NPEFF operates on the hypothesis that a neural network's processing of inputs can be decomposed into a hierarchical set of abstract sub-computations. The computation performed by a model therefore makes use of a fixed set of learned sub-computations, and its processing on any particular example involves a sparse subset of them. Whether a given sub-computation is used for a given example therefore depends on the abstract concepts that underlie the example. Across examples, we assume that each sub-computation will involve a consistent subset of parameters that perform a similar operation. Since the Fisher information matrix for each example relates perturbations in model parameters to changes in the model's predictive distribution, the sub-computations applied to an example become imprinted in it.

NPEFF disentangles these sub-computations given a collection of per-example Fishers (PEFs) through a decomposition procedure. We provide two versions of NPEFF that operate on different representations of the PEF matrices: a diagonal approximation and an exact low-rank representation. When using diagonal PEFs, our decomposition procedure becomes equivalent to non-negative matrix factorization (NMF) (Lee & Seung, 1999). In the low rank case, we represent each example's PEF matrix as a non-negative sum of rank-1 positive semi-definite matrices. To the best of our knowledge, this is a novel decomposition, and we introduce a scalable, multi-GPU algorithm for computing it.

The output of an NEPFF decomposition provides a direct way to quantify the importance of each component to the model's processing of each example. Looking at the examples most influenced by a particular component provides a means to infer what concept or heuristic the component represents. Furthermore, NPEFF generates a "pseudo-Fisher" matrix for each component that can be used to estimate the impact of parameter perturbations on their associated sub-computations. By constructing perturbations that selectively impact particular components, we can verify that the components reflect how the model actually processes examples.

Overall, NPEFF provides a way to interpret a model's processing by extracting the computational sub-steps it uses. In our experiments on text and vision models, we find that these sub-steps often corresponded to human-recognizable concepts by inspecting component top examples. Our parameter perturbation experiments demonstrate that the component pseudo-Fishers indeed reflect parameters important for their particular computational steps. We further ran experiments on a TRACR-compiled toy model implementing a known ground-truth algorithm and demonstrate that components align with sub-steps of the computation. We also explored a potential application of NPEFF's parameter perturbation to correcting faulty heuristics used by a model. Finally, we include experiments demonstrating the robustness of NPEFF to choices of hyperparameters.

## 2 NON-NEGATIVE PER-EXAMPLE FISHER FACTORIZATION (NPEFF)

### 2.1 FISHER INFORMATION

Consider a classification model $p_\theta(y|\mathbf{x})$ with parameters $\theta \in \mathbb{R}^m$ that maps inputs $\mathbf{x} \in \mathcal{X}$ to a softmax distribution over $C$ labels. Given any example $\mathbf{x} \in \mathcal{X}$, we define the per-example Fisher (PEF) matrix as

$$F(\mathbf{x}) = \mathbb{E}_{y \sim p_\theta(y|\mathbf{x})} \nabla_\theta \log p_\theta(y|\mathbf{x}) \nabla_\theta \log p_\theta(y|\mathbf{x})^T. \tag{1}$$

The Fisher information matrix has an information geometric interpretation as a metric relating local perturbations in parameters to changes in the model's predictive distribution (Amari, 2016)

$$D_{\mathrm{KL}}(p_\theta(y|\mathbf{x}) \| p_{\theta+\delta}(y|\mathbf{x})) \approx \frac{1}{2} \delta^T F(\mathbf{x}) \delta \tag{2}$$

as $\delta \to \mathbf{0}$, where $D_{\mathrm{KL}}$ is the KL divergence.

If we let $\mathbf{a}_j(\mathbf{x}) = \sqrt{p_\theta(y_j|\mathbf{x})} \nabla_\theta \log p_\theta(y_j|\mathbf{x})$, then we can express the PEF as $F(\mathbf{x}) = \sum_{j=1}^C \mathbf{a}_j(\mathbf{x}) \mathbf{a}_j(\mathbf{x})^T$. We can thus represent the full PEF matrix using only $Cm$ values, which we call its low-rank representation or LRM-PEF. Alternatively, we can use the diagonal of the PEF matrix as its representation, which we call the diagonal PEF or D-PEF (Kirkpatrick et al., 2017). In this case we have

$$\mathbf{f}(\mathbf{x}) = \mathbb{E}_{y \sim p_\theta(y|\mathbf{x})} \left( \nabla_\theta \log p_\theta(y|\mathbf{x}) \right)^2. \tag{3}$$

Unlike the LRM-PEF which is an exact representation, the D-PEF corresponds to the approximation $F(\mathbf{x}) \approx \mathrm{Diag}(\mathbf{f}(\mathbf{x}))$. Generally, D-PEFs are more tractable to process than LRM-PEFs when the number of classes $C$ is large.

**Sparsity** Since the number of parameters $m$ can be very large for real world models, storing D-PEFs or LRM-PEFs for a modestly sized data set can become intractable. Fortunately, we empirically observe that most of the entries of the PEFs for typical trained neural networks tend to be very small in magnitude. This is to be expected from prior works on model pruning (Hoefler et al., 2021), which finds that most parameters are not important for the model's behavior (Frankle & Carbin, 2019). In our work, we fix some value $K \in \mathbb{N}$ and sparsify each PEF representation by using only the $K$ values with the largest magnitudes. This significantly reduces the amount of required storage and compute with relatively little impact on the accuracy of the representations.

**Number of Classes** For tasks with only a few classes, we can include the term for every class in equation 1 and equation 3 when computing the PEFs. However, this becomes prohibitively expensive as the number of classes increases, requiring roughly the same amount of computation as required for a backwards pass for each class. For such tasks, we thus discard terms correspond to classes whose probabilities $p_\theta(y|\mathbf{x})$ are below some threshold $\epsilon$.

## 2.2 DECOMPOSITION

Let $\mathcal{D} = \{\mathbf{x}_1, \ldots, \mathbf{x}_n\}$ be a set of examples. Let $F_i$ correspond to our representation of the Fisher matrix for the $i$-th example, i.e. $F_i = \sum_{j=1}^C \mathbf{a}_j(\mathbf{x}_i)\mathbf{a}_j(\mathbf{x}_i)^T$ if we are using LRM-PEFs or $F_i = \text{Diag}(\mathbf{f}(\mathbf{x}_i))$ for D-PEFs. The decomposition central to NPEFF can be expressed as the non-convex optimization problem

$$
\begin{aligned}
\text{minimize} \quad & \sum_{i=1}^n \|F_i - \sum_{j=1}^r W_{ij} H_j\|_F^2 \\
\text{subject to} \quad & W_{ij} \geq 0, \\
& H_j \in \mathcal{H},
\end{aligned}
\tag{4}
$$

where $\mathcal{H}$ corresponds to a subset of $m \times m$ matrices and $r$ is a hyperparameter denoting the number of components to learn. We choose this decomposition since we can interpret the $H_j$ as a "pseudo-Fisher" for the $j$-th component, and the non-negative coefficients $W_{ij}$ allow us to see the contributions of each component to the KL-divergence of a model's predictions following a perturbation of its parameters.

For LRM-NPEFF, $\mathcal{H}$ is the set of rank-1 $m \times m$-positive semi-definite (PSD) matrices. Any element of this $\mathcal{H}_j$ can be expressed as $H_j = \mathbf{g}_j\mathbf{g}_j^T$ for some vector $\mathbf{g}_j \in \mathbb{R}^m$. This decomposition can be described as approximating a set of low-rank PSD matrices as non-negative combinations of a set of shared rank-1 PSD matrices. We present a multi-GPU algorithm performing this decomposition in appendix A. Importantly, this algorithm does not explicitly construct any $m \times m$-matrix and instead uses a number of inner products between $m$-dimensional vectors that is independent of $m$ itself.

For D-NPEFF, $\mathcal{H}$ is the set of diagonal $m \times m$-matrices with non-negative entries. Any element of this $\mathcal{H}$ can be expressed as $H_j = \text{Diag}(\mathbf{h}_j)$ for some non-negative vector $\mathbf{h}_j \in \mathbb{R}_{\geq 0}^m$. Solving equation 4 then reduces to the well-studied problem of non-negative matrix factorization (NMF) (Wang & Zhang, 2012). We used a multi-GPU implementation based on Boureima et al. (2022).

Both variants of NPEFF produce a non-negative matrix $W \in \mathbb{R}_{\geq 0}^{n \times r}$ that we call the coefficient matrix. It defines the relationship between individual examples and components. Each entry $W_{ij}$ represents the contribution of the $j$-th component to the PEF of the $i$-th example. The coefficient matrix allows us to get a qualitative understanding of the sub-computation associated to each component. Given a component, we create a list of the examples sorted by the component's coefficient and look at the ones with the highest coefficients. Those top examples often display an interpretable pattern from which we can infer what processing the component represents.

Both NPEFF variants also produce a collection of $m$-dimensional vectors that we refer to as the component pseudo-Fishers. Deferring to section 2.3 for further exposition, each component's pseudo-Fisher vector generates what can be interpreted as an analog of a Fisher information matrix for that component. We express these matrices as $H_j = \mathbf{g}_j\mathbf{g}_j^T$ for an LRM-NPEFF pseudo-Fisher vector $\mathbf{g}_j \in \mathbb{R}^m$. D-NPEFF generates $H_j = \text{Diag}(\mathbf{h}_j)$ for the non-negative pseudo-Fisher vectors $\mathbf{h}_j \in \mathbb{R}_{\geq 0}^m$.

**Preprocessing** During initial experimentation, we found that using raw PEFs led to components tuned to outlier examples. We suspect that outlier examples tend to have PEFs with large magnitudes, and thus their contributions dominate the loss during optimization. To rectify this, we normalized PEFs to have unit Frobenius norm *before* sparsification. This has the effect of de-emphasizing examples for which the sparse approximation is poor; however, we expect any difference in the outcome to be minimal. Efficient computation of the Frobenius norm is described in appendix B.

Our decomposition algorithms make use of a dense representation of the pseudo-Fisher vectors, and thus a naive implementation would require the use of $rm$ floats to represent them. However, the sparsification of the PEFs leads to many parameters having few non-zero corresponding entries. We are able to greatly reduce the memory burden by pruning these entries across the PEFs and using a

dimensionality of $m' < m$. We suspect these positions do not contain enough effective "data-points" to contribute meaningfully to the factorization, so its validity should be minimally impacted.

**Coefficient Fitting** Once we have computed the pseudo-Fisher vectors on a data set $\mathcal{D}$, we can fit a coefficient matrix to PEFs created from another data set $\mathcal{D}'$. Importantly, this allows us to see if the tunings of NPEFF components generalize to examples not used to generate them. Since both of the LRM-NPEFF and D-NPEFF decomposition algorithms operate by alternating between updating the coefficient matrix and updating the pseudo-Fisher vectors, we can fit coefficients to a fixed set of pseudo-Fishers by repeatedly only performing the coefficient matrix update step. By caching intermediates that do not depend on the coefficients, coefficient fitting can be performed far more efficiently than a full NPEFF decomposition.

**Components Set Expansion** Having learned an NPEFF decomposition using a large, general set of examples, we can expand the set of components to create components specialized to another set of examples that are of particular interest. The process is similar to that of a regular NPEFF decomposition with the exception that a subset of component pseudo-Fishers are initialized using the existing decomposition. These are then frozen throughout the decomposition procedure. This leads to the frozen components capturing the general-purpose processing trends leaving the non-frozen components to capture processing specific to that set of examples.

### 2.3 Guided Perturbations

Recall from equation 2 how PEF matrices can be used to relate perturbations of model parameters to changes in its predictive distribution for an example. Using the NPEFF decomposition, we can approximate a PEF matrix for example $\mathbf{x}_i$ as $F_i \approx \alpha_i \sum_{j=1}^{r} W_{ij} H_j$, where $\alpha_i$ is the Frobenious norm of $F_i$. Plugging this into equation 2 gives us

$$D_{\mathrm{KL}}(p_\theta(y|\mathbf{x}_i)\|p_{\theta+\delta}(y|\mathbf{x}_i)) \approx \frac{\alpha_i}{2} \sum_{j=1}^{r} W_{ij} \delta^T H_j \delta \tag{5}$$

for a parameter perturbation $\delta \in \mathbb{R}^m$. From this, our choice of calling $H_j$ the "pseudo-Fisher matrix" for the $j$-component should be clear: it can be used to relate the disruption of the model's predictions on examples utilizing the component to perturbations of parameters. Furthermore, this justifies our choice of decomposition method since the uncovered factors can be readily interpreted in equation 5. This provides a theoretically principled way to verify relationship between parameters and examples uncovered by NPEFF. The goal is to find a perturbation $\delta \in \mathbb{R}^m$ such that $\delta^T H_k \delta$ is large when $k = j$ for a chosen component $j$ and is small otherwise. Once one is found, we can compare the KL-divergences between the perturbed and original models' predictions on examples to their coefficients for the $j$-th component.

For LRM-NPEFF, recall that each pseudo-Fisher matrix $H_k$ can be expressed as $\mathbf{g}_k \mathbf{g}_k^T$ for some vector $\mathbf{g}_k \in \mathbb{R}^m$. Thus $\delta^T H_k \delta = (\mathbf{g}_k^T \delta)^2$. We wish to find a $\delta \in \mathbb{R}^m$ such that $\mathbf{g}_k^T \delta$ has a large magnitude for $k = j$ and small magnitude otherwise. We found that simply using $\delta \propto \pm \mathbf{g}_j$ performed well, but the selectivity of the perturbation could be improved by taking the orthogonal rejection of $\pm \mathbf{g}_j$ onto a subset of the other components' pseudo-Fishers. Namely, we orthogonally reject $\pm \mathbf{g}_j$ onto $\mathbf{g}_k$ if the magnitude of their cosine-similarities is below some threshold. The threshold is used to prevent similar components from interfering in the construction of the perturbation. We found a threshold of 0.35 to work well. The construction of the perturbation for D-NPEFF is more involved and is presented in appendix C.

## 3 Experiments

### 3.1 Component Tunings and Perturbations

We ran LRM-NPEFF on two NLP models and ran D-NPEFF on a vision model. The NLP models were BERT-base fine-tuned on MNLI and QQP, respectively (Devlin et al., 2018). QQP is a binary classification task that aims to determine whether a pair of questions are duplicates of each other (Wang et al., 2018a). MNLI is a natural language inference (NLI) task that involves determining

whether a given hypothesis is implied by, contradicted by, or neutral with respect to a given hypothesis (Williams et al., 2018). Thus, it has three classes. For the vision model, we used a ResNet-50 (He et al., 2016) trained on the ImageNet classification task (Russakovsky et al., 2015). ImageNet involves classifying an image as one of 1000 fine-grained classes.

When learning the NPEFF components for QQP, we used 50k examples heldout from the canonical train split to produce PEFs. For the NLI model, we used 50k examples from the SNLI data set (Bowman et al., 2015). Like the MNLI data set used to train the model, SNLI is also an NLI task with the main difference being that SNLI examples come from a single domain while MNLI has examples from multiple domains. We chose to do this to get a better idea of how NPEFF uncovers heuristics when a model generalizes to new data sets. We used 20k examples from the train split of ImageNet 2012 to produce PEFs for the vision model. We ignored classes with a probability of less than 3e-3 when computing the PEFs for the vision model.

We used 512 components when running NPEFF on the NLI and vision models, and we used 256 components for the QQP model. Once the NPEFF pseudo-Fishers were computed, we fit coefficients to PEFs from a heldout set of examples for each model. We used 50k SNLI examples for the NLI model, 40,430 examples from the QQP validation set for the QQP model, and 30k examples from the ImageNet validation set for the vision model. All of the component tunings and perturbations presented in this section are from these heldout sets. More details on the models and data sets used can be found in see appendix D. There, we also go into detail about the hyperparameters used when computing the PEFs and running NPEFF for these experiments.

### 3.1.1 COMPONENT TUNINGS

COMPONENT 70:
```
Which is the best dish tv connection in hyderabad?
Which is the best dish tv connection in bangalore?

Which is the best eye hospital in kolkata?
Which is the best eye hospital in pune?

Where can i buy second hand books in hyderabad?
Where can i buy second hand books in india?

What is the best coaching center for gre in chennai?
What is the best coaching center for gre in bangalore?
```

COMPONENT 345:
```
[P] Two men in the kitchen with a pot on the stove.
[H] Two men cook a pot of soup in the kitchen.

[P] A young girl is eating as she sits at a table full
of food.
[H] A young girl is eating fruit at a table.

[P] A child is looking at an exhibit.
[H] A child looks at an exhibit of a monkey.

[P] 3 men are working on a boat.
[H] 3 men are working on a crab boat in alaska.
```

Figure 1: Top examples for two language components. The examples presented are the ones with the 4 highest coefficients. **(Left)** QQP component. The model correctly predicts that these questions pairs are not duplicates. Despite asking for the same information, they differ in geographic location. **(Right)** The model correctly predicted "neutral" for these examples. The hypothesis is consistent with the premise but includes additional information that cannot be inferred from the premise.

COMPONENT 36

COMPONENT 218

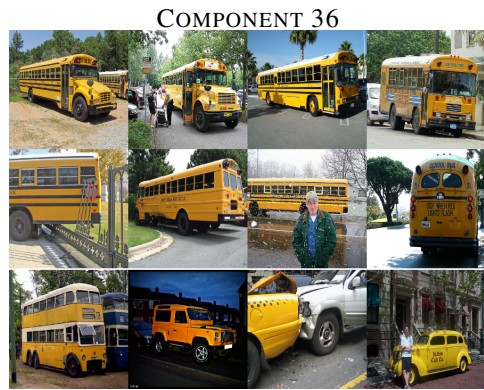
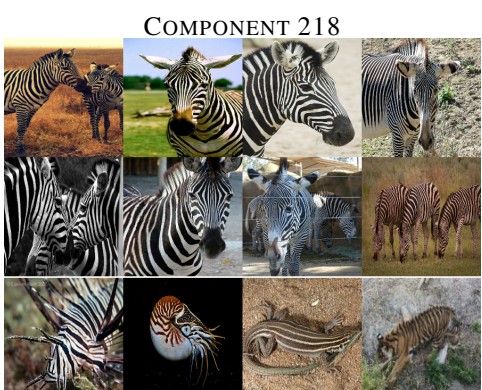

Figure 2: Top examples for two vision components. The top two rows for each component are the examples with the 8 highest coefficients. The bottom row for each component contains selected examples from the set of top 32 examples. **(Left)** The examples with the highest coefficients are all school buses. In general, examples with high coefficients have a yellow vehicle in them. **(Right)** The examples with the highest coefficients are all zebras. In general, examples with high coefficients have a striped animal in them. The specific type of animal does not seem important with fish, mollusks, lizards, and mammals being included.

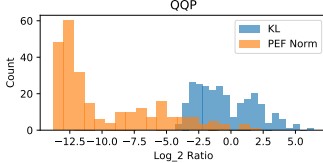 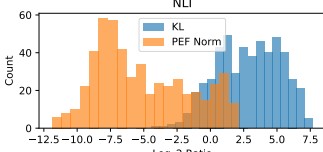 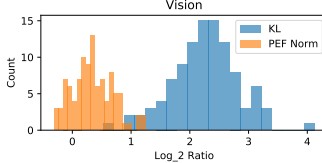

Figure 3: Histograms of the $\log_2$ ratios between top component examples and random examples for KL-divergence and PEF norm.

For all models, we found that most components appeared to have some potential interpretable tuning. We present the top examples of some components in fig. 1 for the NLP models and fig. 2 for the vision model. More examples of tuned component top examples are provided in appendix J.

Some components for the NLP models appeared to be tuned to strategies of solving their respective tasks for specific types of examples. These were typically selective for one of the predictions, and the top examples all contained some common task-specific group of features. Other components appeared to be tuned for features not directly related to the task (e.g. existence of certain kinds of words). It was often the case that a component would use a combination of these two tuning styles.

Many components for the vision model had tunings that included relatively low-level features such as color, shape, and spatial patterns. More abstract, high-level features also played in a role in the tunings of many components. These range from broad, coarse-grained attributes such as the general category of the image subject (e.g. animal, plant, vehicle, architecture, technology, etc.) to more specific, fine-grained attributes such as the fur patterns of dogs. Many of the components had tunings that included both low-level and high-level features. In those cases, the low-level features were often predictive of the higher level ones.

### 3.1.2 PERTURBATIONS

For the LRM-NPEFF experiments on the NLI and QQP models, we used the perturbation method discussed in section 2.3. Perturbation vectors were scaled to have an L2 norm of 0.1 before applying them to the model parameters. We then measured the per-example KL-divergence of the perturbed model's predictions from the original model's predictions. To score the selectivity of the perturbation's impact on the top examples of a component, we calculated the ratio of the average of this KL-divergence for the component's top 128 examples to the average KL-divergence for a random subset of 10k examples. Since the sign of the perturbation is not determined by the method in section 2.3, we try both adding and subtracting the perturbation to the model parameters. The reported scores correspond to whichever has a higher KL-ratio. Details of the perturbation experiments for the D-NPEFF decomposition of the vision model can be found in appendix E.

Looking at the expression equation 5 relating the KL-divergence of a model's prediction to a perturbation of its parameters, we see that the KL-divergence is directly proportional to norm of the example's PEF. In contrast, the coefficients used to rank the top examples for a component are computed using normalized PEFs and thus are independent of their norms. To explore the impact of this confounding factor, we computed the ratio of the average PEF Frobenius norm of the top 128 examples for each component to the average norm of a random subset of 10k examples.

Histograms of the KL-divergence and PEF norm ratios are presented in fig. 3. For all models, these histograms clearly show that the KL-divergence ratios tended to be significantly higher than the PEF norm ratios. Hence the perturbations crafted using a component's pseudo-Fisher had a larger impact on the model's processing of the component's top examples than would be expected solely from their general sensitivities to parameter perturbations. This supports our claim that the directions in parameter space uncovered by NPEFF are specifically important for their respective component top examples. Furthermore, this helps verify that the concepts and heuristics uncovered by NPEFF correspond to actual processing methods used by the model.

## 3.2 TOY MODEL WITH KNOWN GROUND TRUTH

Since it is difficult to evaluate whether a method uncovers concepts actually used by a real-world model, we ran experiments using a synthetic task and transformer model compiled by TRACR (Lindner et al., 2023) that implements a RASP program (Weiss et al., 2021) This task simulates an extremely simplified NLI task, and its detailed description and RASP program solving it is provided in appendix H. The RASP program is compiled by TRACR to an encoder-style Transformer (Vaswani et al., 2017) with 4 attention heads, a model dimension of 25, feedforward hidden dimension of 1320, and 23 layers with 23.5M params.

We ran LRM-NPEFF on 1616 examples using either 32 or 128 components. Since we know how the model processes examples, we can programmatically search for components whose top examples all contain a concept used by the model. Namely, we use the values of Boolean intermediate variables in the RASP program to indicate whether an example contains a concept. The meaning of a concept can be ascertained by looking at the role of its associate variable in the human-readable RASP program. We found that 30/32 and 92/128 components had 128 top examples all containing at least one concept. The overall set of concepts found in the components was diverse. Many of the components were to tuned to the logical AND of multiple intermediate RASP variables, which can be seen as being tuned to more fine-grained concepts. See appendix H for details on this programmatic search along with a detailed breakdown of component tunings. Although caution should be used when extending these results to real-world models since organic processing might differ from TRACR's implementations, these results demonstrate that NPEFF is able to extract concepts actually used by a model in this setting.

## 3.3 HYPERPARAMETER EXPLORATION

The NLI model's LRM-NPEFF from section 3.1 was used as the baseline setting for hyperparameter exploration. To concisely compare component tunings, we used the cosine similarity of component coefficient vectors. Components with a high cosine similarity tended to have similar top examples.

**Sparsity**    We ran LRM-NPEFF keeping the top 16384, 65536, and 262144 values per PEF. We used 20k examples and 128 components. Given a pair of runs, we looked at each component from one of the runs and recorded its max cosine similarity with any component from the other run. If this value is high for most components in both directions for a pair of runs, then most of the concepts from one run have an analogous concept in the other. This metric had an average value of 0.77 when comparing the components from the 16384-value run against either of the other runs. The 65536-value run had an average of value of 0.76 and 0.80 when compared against 16384-value and 262144-value runs. The 262144-value run had an average of value of 0.77 and 0.80 when compared against 16384-value and 65536-value runs. Hence the sets of learned components tended to be fairly similar with no major qualitative difference being observed between their tunings. Information about how well the PEFs were approximated at different levels of sparsity can be found in appendix G.

**Number of Components**    We ran LRM-NPEFF using 32, 128, and 512 components. To compare a pair of runs A and B, suppose that run B has more components than run A. For each component $i$ of run A, we found the subset of run B's components whose coefficient cosine similarity to that component $i$ was greater than to any other of run A's components. For all pairs of runs, we found that every component from the smaller run had at least one corresponding component from the larger run. Qualitatively, we found that the groups of components from the larger run corresponding to each component from the smaller run had similar tunings. More precisely, these groups tended to be more fine-grained "splits" of the component from the smaller run. For example, a component tuned to hypotheses containing a word that contradicts a word in the premise might have a split where the relevant words are colors and another split where they are "cat" and "dog".

**Number of Steps**    To explore the convergence of the coefficient matrix when running NPEFF, we performed the decomposition for 1500 steps and saved the matrix every 50 steps. A graph of the average component coefficient cosine similarity with its final value across steps can be found in fig. 5. Notably, the average similarity exceeded 95% by 700 steps and 99% by 1100 steps.

**D-NPEFF vs. LRM-NPEFF**    We repeated the experiments of section 3.1 for the NLI and vision model but using D-NPEFF instead of LRM-NPEFF and vice-versa. For the NLI model, we found D-NPEFF to generate fewer components with readily interpretable tunings than LRM-NPEFF. While most of the LRM-NPEFF components appeared tuned, only about half of the D-NPEFF components appeared tuned. Furthermore, the perturbation of D-NPEFF components led to less selective changes in model predictions than for the LRM-NPEFF components with a median KL-ratio of 5.26 compared to 7.64. However, this might just be an artifact of the difference between the perturbation methods of the two flavors of NPEFF.

The LRM-NPEFF results on the vision model were far worse than the D-NPEFF results. Far fewer components had interpretable tunings, and top examples of tuned components were significantly noisier. The selectivity of perturbations of the LRM-NPEFF components was also lower than for the D-NPEFF components with a median KL ratio of 2.45 compared to 4.81. We suspect that using a fixed number of non-zero entries led to examples with high rank PEFs having poor sparse approximations. Left to future work, possible methods to adapt LRM-NPEFF to tasks with many classes (and thus potentially high rank PEFs) include varying the number of kept entries with PEF rank and using a different low-rank approximation to the PEF instead of just using a subset of classes.

### 3.4    Comparison to Existing Interpretability Methods

While there are no existing unsupervised concept discovery algorithms directly comparable to NPEFF, we can adapt the methods of Ghorbani et al. (2019) to be a baseline. Ghorbani et al. (2019) learns concepts through k-means clustering of activation space representations of image segments. We compared NPEFF to a more generalized version of this operating directly on examples rather than image segments, which allowed us to run experiments on text tasks. We used the distance of an example's activations from its cluster's centroid to rank examples within a cluster.

We ran this baseline for both the NLI and vision models. We used the encoder's final representation of the `CLS` token as the activations for the NLI model. For the vision model, we used the output of the global average pooling layer that immediately precedes the classification layer. Using the same sets of examples as in section 3.1, we learned 128 clusters for each model. Most of the top examples for the clusters displayed some human-interpretable tuning as ascertained through examination of their top examples. Top examples of some clusters are presented in appendix M. We used TCAV (Kim et al., 2018) to test whether a relationship existed between the top examples for a cluster/component and the prediction of at least one class for the NLI model. We found a statistically significant relationship existed for every k-means cluster and every LRM-NPEFF component. Details can be found in appendix I.

Qualitatively, we found some similarities in the tunings of NPEFF and baseline components with NPEFF producing a more diverse set of tunings. To get a more quantitative comparison, we used the average max cosine-similarity metric from section 3.3 to compare tunings. Here, we used a vector of 0s and 1s indicating cluster membership as the "coefficients vector" for a k-means cluster. For the NLI model, we found that k-means clusters had an average max cosine similarity of 0.20 against the NPEFF components and a value of 0.19 in the opposite direction. Since NPEFF coefficients and cluster "coefficients" represent different types of quantities, we should note that this metric provides a cruder similarity measure than when comparing between NPEFF components.

Nevertheless, we would expect differences in recovered concepts. The differences in recovered concepts highlights several advantages of NPEFF. The k-means baseline operates on activations from a single layer near the output of the model while NPEFF operates on PEFs containing information from the entire model. Hence NPEFF can pinpoint what part of the model performs certain computations. In contrast, a full-model decomposition over activations becomes tricky when the total activation space does not have fixed dimension (e.g. the variable sequence length of transformers). The ability to make informed modifications to parameters to directly test whether the model actually uses a recovered concept is a further advantage unique to NPEFF.

### 3.5    Example Application: Fixing Flawed Heuristics

We experimented using NPEFF with the NLI model to see what heuristics it uses when it makes incorrect predictions. To do so, we created a filtered set of PEFs corresponding to examples on

which the model made an incorrect prediction. We then performed the components set expansion procedure to learn 64 components specialized to these examples. Finally, we fit coefficients to a non-filtered set of examples using the expanded NPEFFs. See appendix F for details on this process.

The components in the expansion were more likely to have a high fraction of incorrect predictions in their top examples. More precisely, 41% of the expanded components had incorrect predictions for at least 50% of their top 16 examples compared to just 4.3% of the original components. Such components could generally be placed into one of three groups: correct heuristics with inconsistent labeling, completely wrong heuristics, or flawed heuristics. We can try to alter the model parameters to selectively fix components tuned to incorrect heuristics. We selected the subset of expanded components whose top 16 examples contained at least 8 incorrect predictions. The top examples of some these components are provided in appendix L. We then performed perturbations on these components using a similar methodology to section 3.1.2 but with a larger perturbation magnitude of 3e-1. Out of the 48 components deemed to correspond to incorrect heuristics, we found four components that increased the model's accuracy by at least 0.5% after being perturbed. While these results are promising, we emphasize that this method of fixing faulty heuristics using NPEFF is mostly a proof of concept with improvements such as using loss gradient information left to future work.

## 4 RELATED WORK

Methods reliant on ranking or grouping examples are common in the interpretability literature. For example, the activation value of a single neuron is a frequently used metric to derive such rankings (Sajjad et al., 2022; Bolukbasi et al., 2021; Bau et al., 2017; Szegedy et al., 2013). Alternate methods include using agglomerative clustering on token-level BERT representations to find groups of representations corresponding to concepts (Dalvi et al., 2022), using supervised probing task to learn latent interpretable representations (Michael et al., 2020), and unsupervisedly finding interpretable directions in the latent space of a generative adversarial network (Voynov & Babenko, 2020).

Kim et al. (2018) uses user-defined groups of examples to represent an abstract concept. Ghorbani et al. (2019) uses clustering of activation space representations of image segments to automatically learn concepts. Yeh et al. (2020) learns concepts by projecting activations at a given layer onto a set of learned vectors, setting the dot products below a threshold to zero, and then using a learned mapping back to activation space before being processed by the rest of the network. They add regularizers to encourage concepts to be coherent and unique.

Representing a model as a set of sub-computations underlies some vision interpretability work. Zhang et al. (2018) learns an explanatory graph given a convolutional neural network. They create nodes corresponding to disentangled convolutional filters and assign edges based on the relative spatial displacement of node activations between neighboring layers. For each example, Wang et al. (2018b) learns a scalar control gate for each channel in each layer of a convolutional network. They are fit using a loss that encourages the gates to be as sparse as possible while preserving the model's predictions on the example. Performing clustering on these reflects input layout patterns.

## 5 CONCLUSION

NPEFF presents a novel direction in model interpretability research. Applicable to any model with differentiable parameters, it automatically discovers concepts and heuristics used by a model. NPEFF further grounds these in the model's parameter space and allows for their selective perturbations. In addition to being useful in verifying that these concepts and heuristics faithfully represent the model's processing, it opens up the possibility of informed model modifications.

In future work, we hope to further develop some of the applications of NPEFF explored in this paper as well as explore its viability in facilitating scientific discovery. Furthermore, we aim to address some limitations of the current version of NPEFF. In particular, we plan to explore methods to handle models with a large output space such as autoregressive models when computing PEFs such as sampling from the output distribution. We also plan to explore modifications to allow NPEFF to produce components with user-specified tunings, which would facilitate using NPEFF to make user-guided changes to a model.

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
