# OpenReview forum: "NPEFF: Non-Negative Per-Example Fisher Factorization"
_ICLR.cc/2024/Conference — Submitted to ICLR 2024_

### Official Review · Reviewer_Brpd · 2023-10-17

**Soundness:** 3 good
**Presentation:** 2 fair
**Contribution:** 3 good
**Rating:** 6
**Confidence:** 3

**Summary:**

This paper proposes an interpretability method which allows to obtain representations of concepts. These representations are found by non-negative per-example fisher factorization, which can be done for any end-to-end differentiable model. This method is analogous to non-negative matrix factorization matrices in one of its instantiations and hence decomposes the per-example fisher information matrix into a set of non-negative coefficients and concept vectors, referred to as pseudo-Fishers.
In their experiments, the authors demonstrate the ability of their method to verify what concepts lead to the model's processing. They also show some initial experiments demonstrating how to selectively fix incorrect predictions.

**Strengths:**

- The authors propose a novel, low-rank decomposition for the per-example fisher.
- The method is applicable to any end-to-end differentiable model.
- The authors did a thorough analysis of the hyperparameters introduced by their method.

**Weaknesses:**

- While the advantages of the method are well explained, it would be good to add a separate limitations section for transparency. Could you elaborate in your comments on what you see as the biggest limitations.
- I have found the toy model difficult to understand. I think it would be good to provide an intuitive explanation in the main text on the programmatic search.

**Questions:**

- Could your example about fixing flawed heuristics have applications to the unlearning literature?
- As pointed out by Kunstner et al. 2019, the fisher information matrix is an overloaded object which may or may not refer to the empirical fisher. Since you switch from the true fisher to an approximation which Kunstner et al. 2019, please check section 3.2 of their paper that you followed the correct terminology


Typos / Small errors
- Please review your references, e.g. the lottery ticket hypothesis paper appeared in ICLR 2018.
- The equation equation 5 provides ...


References:
Kunstner, Frederik, Philipp Hennig, and Lukas Balles. "Limitations of the empirical fisher approximation for natural gradient descent." Advances in neural information processing systems 32 (2019).

---

> ### Author Response · Authors · 2023-11-18
>
> Thank you for your review.
>
> > Would you elaborate in your comments on what you see as the biggest limitations.
>
> We've added a limitations/future work paragraph to the end of the conclusion. The two main limitations
> we see in our current implementation of NPEFF are supporting models with a large output space (e.g.
> having many classes or text generation) and allowing for control over component tunings.
>
> For the former, we can sample from the output distribution to create an approximation of the PEFs.
> This would require things like determining the number of samples needed per example. Alternatively if
> canonical input/output pairs are provided, we can use something like the empirical Fisher (as per Kunstner et al. 2019)
> instead.
>
> Although the unsupervised nature of component recovery using NPEFF can be considered one of its advantages, it
> can also be a limitation when trying to use NPEFF in some applications. For example, being able to create a
> component tuned for a particular user-defined concept would be useful when using something like our
> perturbation procedure to make user-guided changes to a model. One potential method for this is to create a component whose
> coefficients would be forced to be non-zero for a selected set of examples and zero for all others.
>
>
> > I have found the toy model difficult to understand.
>
> There was general confusion amongst the reviewers about this. We have updated Section 3.2 to more clearly describe
> what the "concepts" were (values of Boolean intermediate variables in the RASP program) and the tunings of the
> recovered components.
>
> > Could your example about fixing flawed heuristics have applications to the unlearning literature?
>
> Possibly. One limitation of NPEFF, however, is the inability to directly create a component tuned for a
> human-defined concept. This could make it challenging to obtain a component tuned for something that you
> want to unlearn. As was done in the flawed heuristics experiments, the components set expansion procedure
> could help in creating components tuned to user-defined concepts/information, but there would be no guarantees
> in actually recovering such a component. However, it is possible that modifications could be made to the
> NPEFF decomposition to procedure to better support this. For example, one could create a component whose
> coefficients would be forced to be non-zero for a selected set of examples and zero for all others.
>
> > The fisher information matrix is an overloaded object which may or may not refer to the empirical fisher.
>
> For PEFs (per-example Fishers), there is no expectation over the model inputs $x$, so there is no conceptual difference
> between a "statistics Fisher" and "statistics empirical Fisher". As for the role of
> the expectation over classes "y", we make no use of any label provided with the example and perform
> an expectation over the set of classes.

---

### Official Review · Reviewer_RGwC · 2023-10-17

**Soundness:** 3 good
**Presentation:** 3 good
**Contribution:** 2 fair
**Rating:** 5
**Confidence:** 3

**Summary:**

The authors propose a new method for interpreting neural networks called NPEFF. The method applies to classification models and relies on first constructing the Fisher matrix for each example. This matrix can be sparsified by magnitude clipping and also by omitting classes with low probability. Given the Fisher matrices for many examples, the “components” are extracted by approximately factorizing the Fisher matrix in a fashion close to non-negative matrix factorization. The authors also shows how the extracted matrices can be used to relate changes in model parameters to changes in the outputs. Experimentally the authors considers two settings: image classification with Resnets and NLP tasks with a finetuned BERT model. In Figure 1 and 2 the authors visualize some found components, in figure 3 they also show that the model is sensitive in the directions uncovered by NPEFF. Additionally, a comparison with toy data with known ground truth components is done, some ablation experiments are included and a comparison to a previous interpretability strategy is conducted.

**Strengths:**

Intrepretability is important, especially for LLMs.

The paper is well written.

**Weaknesses:**

Interpretability has been studied for a long time, so the novelty is rather low.

The results are only presented for small Bert/Resnet models. It would be better with results for LLMs.

It is hard to know if the proposed method works well. Interpretability is very subjective, and a few examples (which could be cherry picked) are not very convincing. The authors only compare against a single baseline.

**Questions:**

Could you compare against more baselines?

Could you do some kind of human evaluation? E.g. give a sample of 10 people two interpretability models and ask which one they prefer.

Could you give results on LLMs?

---

> ### Author Response · Authors · 2023-11-18
>
> Thank you for your review.
>
> > Interpretability has been studied for a long time, so the novelty is rather low.
> > ...
> > Could you compare against more baselines?
>
> While interpretability has been studied for a long time, we argue that it is a broad enough field
> that there still exists room for significant novelty. In particular, unsupervised concept discovery
> methods like NPEFF are highly novel. We were only able to find two existing unsupervised concept discovery
> methods: Ghorbani et al. (2019) and Yeh et al. (2020). These both operate by mapping activations to concepts.
> For models without a fixed dimension activation space, these methods need some way to get fixed-dimensional (set of) activations
> for example. Neither previous work experimented on Transformers, but you could use token-level activations at a particular layer.
> Since these are not example-level activations, they can leave out important information that is spread out across multiple sequence
> positions. In constrast, NPEFF provides a full example-level decomposition of the model's processing.
>
> In our baselines, we compared NPEFF to Ghorbani et al. (2019) by using activations that come right before the classification
> head. Although this allowed us to get an example level decomposition for the baseline, the activations in the transformer body
> cannot be analyzed this way. Yeh et al. (2020) experimented with performing a sequence/image-position level decomposition of
> activations. However, they only experimented with convolutional models. Some of their methods rely on the properties of the
> receptive fields of intermediate activations in convolutional models, so their method would be harder to adapt to transformers.
>
> > Interpretability is very subjective, and a few examples (which could be cherry picked) are not very convincing.
>
> Succinctly evaluating interpretability methods is tricky due to the generally subjective nature of interpretability.
> We've added to the supplemental material a pdf (mnli_to_snli.pdf) of all of the component top examples for the NLI experiment from Section 3.1.1. While not ideal, these can be used to verify that most components have an interpretable tuning and that the examples included in the paper were not cherry-picked.
>
>
> > Could you do some kind of human evaluation? E.g. give a sample of 10 people two interpretability models and ask which one they prefer.
>
> While human evaluation might be useful for verifying that components have interpretable tunings, we think that using it to compare
> NPEFF to another interpretability model would have many pitfalls. The goal of an unsupervised concept discovery method like NPEFF is to uncover components that accurately reflect the processing done by a model. An interpretability method that produces
> overly simple components might win out in a human evaluation over a model that more accurately reflects the processing done
> by the model. Such comparisons could be of value if the goal of an interpetability method was to get the model to "show
> its work" to an end user, but that was not a goal of NPEFF.
>
> > Could you give results on LLMs?
>
> Our current method of computing PEFs uses an expectation over the set of possible model outputs. For
> classification models with few classes, we compute this expectation exactly, and we ignore classes
> assigned low probabilities for classification models with many classes. For autoregressive language models, however,
> we would need to approximate the PEF by doing something like taking several samples from the model's output.
> If model inputs are paired with ground-truth outputs, then we could use something like empirical Fisher instead of the
> true Fisher. We leave exploration of this to future work.
>
> In terms of applying NPEFF to large models in general (e.g. xlm-roberta-xl and xlm-roberta-xxl), NPEFF doesn't have
> any methodological issues. We can include experiments at that scale, but we do not have the time to run them during
> the rebuttal period.
>
> Operating on sparse representations of PEF matrices allows NPEFF to better scale to large models.
> It is also possible to run NPEFF on PEFs computed using only a subset of the model variables. For example, one might
> run NPEFF using only the parameters from a single transformer layer. While we do not explore that in this work,
> we have explored this a bit and found the components to have interpretable tunings.

---

> > ### Comment · Reviewer_RGwC · 2023-11-22
> > **reply**
> >
> > I thank the authors for their reply. My concerns remain and I will retain my score.
> >
> > I agree that there are pitifalls regarding using humans for evaluation, but maybe it would be possible to recruit a small cohort of DNN researchers to evaluate the interpretability? More quantative evaluation on the interpretability would be good.

---

### Official Review · Reviewer_2rEq · 2023-10-26

**Soundness:** 3 good
**Presentation:** 4 excellent
**Contribution:** 3 good
**Rating:** 6
**Confidence:** 4

**Summary:**

The authors propose NPEFF for unsupervisedly discovering the components being used in a learnt model. NPEFF decomposes each example' Fisher information matrix as a non-negative sum of components, so as to discover a set of r components within the network and a coefficient $W_{ij}$ describing the influence of the j-th component on the i-th example's prediction. NPEFF components are examined for two language and one vision dataset, both by viewing the sets of examples strongly associated with each component and by observing the effects of perturbing on components. Lastly, an experiment is performed to discover whether modifying components associated with incorrect predictions will improve predictive performance, resulting in an accuracy gain of 0.5%.

**Strengths:**

- An unsupervised method for concept discovery that works on general NN would be a useful and significant contribution. While there is some related work as detailed in Section 4, much is focused on vision or does not automatically group features together into sets of discrete concepts (for instance, visualizing CNN convolutions creates human-interpretable patterns as to what the convolutions are picking up, but does not group these patterns together itself)
- The resulting components indeed match to desired, human-recognizable concepts in both the language and vision domains
- Quantitative and qualitative evaluations of the method are both very comprehensive

**Weaknesses:**

- The experiment on fixing flawed heuristics achieves only a slight improvement of 0.5% accuracy, by improving predictions for 4/48 components associated with incorrect heuristics
- Unclear how the perturbations experiment in Section 3.1.2 demonstrates that NPEFF's discovered parameter space directions are important as stated at the end of the section. Would be nice to see more potential application of NPEFF with strong results
- Minor typo: definition of $a_j(x)$ in section 2.1 should have $y_j$, not $y_i$

**Questions:**

- In section 2.2's preprocessing section, you state that raw PEFs gave components tuned to outlier examples. Could NPEFF with raw PEFs be useful for some form of OOD detection or prediction confidence measure?
- Could you give more information on how Section 3.1.2 perturbation experiment demonstrates the claims at the end of this section?
- Figure 3: it seems interesting how in QQP and NLI, the PEF norm ratios take a wider range of values than the KL ones, while in ImageNet the opposite is true. Is this meaningful in any way?
- Could you give more details on what constitutes a "concept" and what components resulted in Section 3.2? For instance, did each component tend to focus on one particular concept and components were quite diverse from each other (as likely desirable), or do many components tend to identify with many different concepts so that components are relatively similar to each other?

---

> ### Author Response · Authors · 2023-11-18
>
> Thank you for your review.
>
> > The experiment on fixing flawed heuristics achieves only a slight improvement of 0.5% accuracy,
>
> We agree that 0.5% is a small boost, but a small boost is to be expected when correcting a single incorrect heuristic
> that is used only on a relatively small subset of examples. The change was significantly bigger when we look at
> model's predictions for the component top examples. The four components that provided a boost of 0.5% or greater to
> the total accuracy had accuracies of 27%, 24%, 58%, and 41% on their top 128 examples using the base model. Their
> respective perturbed models reached accuracies of 88%, 72%, 95%, and 77%, respectively, on the same examples.
>
> We also note that these improvements are for correcting
> a single heuristic at a time. We found that perturbating multiple components at once could boost total accuracy
> further, but we did not explore this beyond some preliminary experiments.
>
> > Could NPEFF with raw PEFs be useful for some form of OOD detection or prediction confidence measure?
>
> Potentially. A simple alternative would be just looking at the norm of a PEF for something like OOD detection.
> NPEFF could provide an advantage over this by having components match up with specific reasons why examples
> are OOD or hard to make good predictions on. For example, such a component from a digit identification task
> could be tuned to ambiguous 4/9 and 5/6 digits.
> Knowing that an example had a high coefficient for such a component could be useful for providing a reason why
> one might not want to place much confidence on the model's prediction for a particular example. We haven't done much
> exploration with NPEFF on raw PEFs, but it is a potential future research detection.
>
> > Could you give more information on how Section 3.1.2 perturbation experiment demonstrates the claims at the end of this section?
>
> We've added some text in the last paragraph of that section to help explain this. The goal of the perturbation
> experiments was to show that the direction in parameter space associated to a component (its pseudo-Fisher) is of particular importance
> to the model's processing of the component's top examples. The perturbation experiment tests this by constructing
> a perturbation derived from a components pseudo-Fisher and seeing how much the predictions for each example change
> as measured by their KL-divergences from their original predictions. Taking the ratio of the average KL-divergence for a component's top
> examples to the average KL-divergence for the data set as a whole measures how much more/less the predictions of the top
> examples changed compared to the rest of the examples. However, the model's predictions for some examples are simply more
> sensivitive to perturbations in general, which is measured by the norm of their PEFs. Hence, if component top examples tended
> to have PEFs with larger norms than average, then looking at the KL-divergence ratios alone would be misleading.
> We therefore also looked at the ratios of the average PEF norm for the component's top examples to average PEF norm for the
> rest of the examples. As the KL-ratios tended to be significantly higher than the PER-norm ratios, the component-derived
> perturbations affected the component's top examples more than can be explained by their general sensitivity to perturbations.
>
>
> > Figure 3: it seems interesting how in QQP and NLI, the PEF norm ratios take a wider range of values than the KL ones, while in ImageNet the opposite is true. Is this meaningful in any way?
>
> We think this might be due to the difference in the number of classes between the tasks. The text tasks have 2 or 3 classes, and the
> model can use some heuristics very confidently to make predictions. This leads to these component top examples having very small
> PEF norms. In contrast, ImageNet is a fine-grained image classification task with a 1000 classes. Most strategies used by the
> model wouldn't be able to confidently make predictions for a single class, so component top examples will tend to have norms
> more in line with average examples. This leads to PEF norm ratios taking on a far larger range for QQP and NLI compared to ImageNet.

---

> > ### Author Response · Authors · 2023-11-18
> >
> > > Could you give more details on what constitutes a "concept" and what components resulted in Section 3.2? For instance, did each component tend to focus on one particular concept and components were quite diverse from each other (as likely desirable), or do many components tend to identify with many different concepts so that components are relatively similar to each other?
> >
> > We have updated Section 3.2 to more clearly describe what the "concepts" were and the tunings of the
> > recovered components. Essentially, concepts corresponded to values of Boolean intermediate variables in the RASP program.
> > Since the RASP program was human written and readable, these meanings of these concepts can be ascertained by looking
> > at the associated variable in the RASP code.
> >
> > We found the set of component tunings to be fairly diverse. We also found that many components were tuned to the logical "and"
> > of multiple concepts, which can be thought of as more fine-grained concepts. Compared to the logical "or" of components mentioned
> > in the latter part of your question, this does not make component tuning relatively similar to each other.

---

> > > ### Comment · Reviewer_2rEq · 2023-11-21
> > >
> > > Thank you for the very detailed response! I will maintain my rating, because I think that NPEFF is promising but the paper could also be improved by expanding the empirical results and potentially investigating possible further uses like OOD detection. However, I am still leaning towards accept.

---

### Official Review · Reviewer_19bz · 2023-11-04

**Soundness:** 2 fair
**Presentation:** 1 poor
**Contribution:** 2 fair
**Rating:** 5
**Confidence:** 4

**Summary:**

The current paper focuses on developing an approach to decomposing a fisher information matrix into interpretable components to understand the behavior of a pretrained model. I found the claims and the methodology itself interesting, albeit reminiscent of prior works on importance/saliency estimation of model parameters in network pruning (Molchanov et al., 2017)---the authors already cite some of these papers.

**Strengths:**

I found the proposed methodology itself interesting, though arguably difficult to implement or work with. The available code can help address this.

**Weaknesses:**

My biggest apprehension at the moment is that the paper's writeup is extremely unclear at times. While the first half of the papers reads fine and I'm able to follow, the motivation for the experiments, their setup, and the very results themselves are quite unclear to me. For example, the authors claim all "directions" unveiled by their method encodes some "concept" used by Tracr for manually performing the tasks discussed therein. It is entirely unclear though what a "concept" means---is it mere addition, a Tracr primitive, a composition of primitives, etc.? Such lack of clarity generally left me confused throughout the paper, such that I was not certain what the implication of any of the experiments was. This made it hard for me to judge the paper.

Practicality: While the authors conduct experiments on a BERT model, the proposed method requires at least linear in number of parameters memory. I'm uncertain of the scalability of this approach, therefore. Of course several methods suffer from this problem, but it will be worth discussing this in the paper.

**Post rebuttals comment:** I appreciate the authors' comments. Having gone through the updated paper, other reviewers' comments, and the rebuttals to them, I do think the paper has improved. I still found the writing unclear at times (e.g., while I get what "tunings" are, I don't understand why the term is suddenly informally defined and then casually used thereafter, making the paper confusing to me at points). I'm increasing my score to 5 however, given the updates and responses.

**Questions:**

See weaknesses.

---

> ### Author Response · Authors · 2023-11-18
>
> We have updated Section 3.2 on the toy TRACR/RASP experiment to more clearly describe what the "concepts" were and the tunings of the
> recovered components. Essentially, concepts corresponded to values of Boolean intermediate variables in the RASP program.
> Since the RASP program was human-written and readable, these meanings of these concepts can be ascertained by looking
> at the associated variable in the RASP code.
> We found the set of components tunings to be fairly diverse. We also found that many components were tuned to the logical "and"
> of multiple concepts, which can be thought of as more fine-grained concepts.
>
> Since real-world models do have an associated RASP program, the rest of the paper uses the term "concept" to refer
> more generally as something the model uses to inform its predictions. This can include concrete features (e.g. mention of food)
> or more abstract properties of the example (e.g. having word in the hypothesis that is in direct contradiction to a word
> in the premise). We'd frequently see components tuned to a combination of these types of concepts (e.g. the premise and hypothesis assigning different colors to an article of clothing).
>
> > the proposed method requires at least linear in number of parameters memory.
>
> As discussed in the "sparsity" paragraph of Section 2.1, we make use of a thresholding operation to create a sparse representation
> of each PEF. The number of non-zero values used in this representation is a user-defined hyperparameter, so the amount
> of memory used to store each PEF ends up being significantly smaller than the number of model parameters. For a 110M
> parameter BERT-base, we found even using 16k non-zero values per PEF to perform well. The number of non-zero values
> needed will likely increase with model size but would still remain significantly smaller than the number of parameters.
> Furthermore, it is also possible to run NPEFF on PEFs computed using only a subset of the model variables. For example, one might
> run NPEFF using only the parameters from a single transformer layer. We found this to be effective in preliminary experiments but left it out of the paper due to space considerations.

---

### Meta-Review · Area_Chair_81zG · 2023-12-06

**Metareview:**

This paper applied the Fisher information to unsupervised concept discovery in NLP and vision models. The author proposed to uncover a set of pseudo-Fisher matrices (or vectors as they are rank-1) that can be used to approximate (in a way that is similar to PCA) the per-sample Fisher information matrix. These pseudo Fisher matrices then are used for model interpretability showing meaningful results in "concept discovery".

Strength:

- This paper successfully applies Fisher information to the interpretability of modern deep-learning scenarios. The methodology has a profound background and is simple to understand. It is well-integrated and the writing is of outstanding quality. The model interpretability of NLP and vision models are important topics and could draw significant community interest.

Weaknesses:

- Clarity. Three of the four reviewers are struggling to clarify the basic definitions like "concept". The authors's response is not satisfactory: they should not assume the readers are already NLP experts and familiar with the latest terminologies. There is a disconnection between the early section and the experiments.

- The introduction of the Fisher information can be enhanced: it lacks intuitions and depth. Is this the pioneering work that bridges the concept of Fisher information into model interpretability and parameter perturbations? From the writing, there is no related literature besides Amari's textbook. The authors can make more efforts in positioning this work into the existing literature on the *application* of Fisher information to deep learning interpretability. Why does the factorization of Fisher information help to discover concepts? Any $f$-divergence, not only the KL, is locally Rao and corresponds to the Fisher information metric. The current introduction in sec. 2 is too brief.

- Clearly, this work should not be regarded as theoretical as there are no analytical results.  As a practical contribution, some details should be better clarified, such as how the pseudo-Fisher is learned in terms of algorithm steps and related storage/computational complexities, and how the sparsification (as raised by some reviewers) and normalization (sec 2.2) affect the interpretability performance. The latter can be argued experimentally.

Some reviewer is concerned that the assessment of the results is subjective and suggests a more   *quantitative* evaluation.

**Justification For Why Not Higher Score:**

The majority of the reviews raised concerns about the clarity of the writing. There is still a noticeable space for improvement to address the weaknesses above.

**Justification For Why Not Lower Score:**

N/A

---

### Decision · Program_Chairs · 2024-01-16

Reject